# Oscillating superflow in multicomponent Bose-Einstein condensates

Angela White[1] and Thomas Busch[1]

[1]*Quantum Systems Unit, Okinawa Institute of Science and Technology
Graduate University, Onna-son, Okinawa 904-0412, Japan.*

Conservation of angular momentum depends on the existence of rotational symmetry. However, even in systems where this symmetry is broken, flipping between angular momentum eigenstates often requires an activation energy. Here we discuss an example of superfluid flow in a toroidal potential, which shows sustained oscillations between two different rotation directions. The energy required to change the direction of rotation is taken out of and temporarily restored into the rotational and intra-component interaction energies of the system.

Atomic Bose–Einstein condensates (BECs) have in recent years provided fruitful testbeds for studies of quantum mechanical superflow. These have ranged from the observation and description of single vortices [1, 2] to the creation of stable Abrikosov lattices [3, 4]. However, as standard harmonic trapping potentials are simply-connected topologies, they do not support stable states with higher order winding numbers. The simplest multiply-connected topologies that allow for this are toroidal geometries, and a number of recent theoretical and experimental studies have focused on investigating aspects of superfluid flow in single and multi-component BECs in such potentials [5–15].

BECs are characterised by a macroscopic condensate wavefunction, $\psi$, and the superfluid flow is inviscid and irrotational, with the condensate velocity, $\mathbf{v} = (\hbar/m)\nabla\phi$, dependent purely on the gradient of the phase, $\phi$. In order for $\psi$ to remain single valued, circulation around a closed contour $\mathcal{C}$ in the condensate has to be quantized in units of $\int_{\mathcal{C}} \mathbf{v} \cdot d\mathbf{l} = h/m$, where $h$ is Plancks constant and $m$ is the boson mass. Single-component BECs in multiply connected toroidal geometries are known to allow for stable currents and are therefore perfect systems for studies of quantized superfluidity.

While this quantization condition seems inherent to the idea of flow in toroidal geometries, it can be broken in two-component superfluids, that are in the immisicible azimuthally phase-separated regime [5, 7]. The flow of these systems then demonstrates emergent classical behaviour, with the domain wall requiring classical solid-body rotation, while the bulk of each condensate rotates as a typical superfluid vortex with a corresponding $1/r$ velocity field. This dichotomy of classical and quantum rotation was shown to be resolved by the appearance of a radial flow at the condensate phase boundary [7].

Bose-Einstein condensates can be finely controlled, and experimentalists are increasingly able to create complicated and intricately shaped trapping potentials [16, 17]. In this work we exploit these new possibilities and investigate superfluid flow around a racetrack potential. While an elliptical potential may at first sight be a simple extension of a toroidal geometry, it promises new and interesting physical features of superfluid flow, as due to a breaking of azimuthal and radial symmetry, angular momentum no longer has to be conserved [18]. In particu-

lar, we investigate the flow of a two-component system in the immiscible regime around an elliptically shaped racetrack potential which provides tighter trapping along the minor elliptical axis. The anisotropic trap width around the racetrack then requires that the length of the domain wall between the two condensate components grows and shrinks during the flow around the ellipse, which leads to a competition between hydrodynamic rotational energy and the intra and inter-component interaction energies.

We show below that two flow regimes exist: unidirectional superfluid flow, arising for inter-component interactions close to the miscible-immiscible phase transition point, and oscillating superflow, when the condensate possess insufficient energy to compensate for the growth in interaction energy necessary to grow the domain wall between the two condensate components. In fact, this behaviour is not restricted to racetrack potentials with anisotropic trapping widths, but will hold for two-component immiscible super-flow in any arbitrary geometry where the length of the domain wall is required to grow (or shorten) due to the flow pattern.

The physics of oscillating superflows draws analogies to the circulation reversal of a vortex in a Bose gas that is trapped in an anisotropic harmonic oscillator potential, where it is known that for strong interactions, circulation reversal is inhibited due to the activation energy required [19, 20]. In contrast, we will show below that in a two-component system the activation energy required for superflow direction reversal can be inherent, and depends on the inter-component interaction.

Two-component condensates with negligible thermal clouds, composed from either two different atomic species [21] or two different spin-states of the same atomic species [22], are well described by a set of coupled Gross–Pitaevskii (GP) equations. In the rotating frame, these coupled GP equations, which we numerically solve for the mean-field wave function $\psi_j$ of component $j$ $(j = 1, 2)$, are given by

$$i\hbar\frac{\partial\psi_j}{\partial t} = \left( -\frac{\hbar^2}{2m}\nabla^2 + V_j + \sum_i^{1,2} Ng_{ji}|\psi_i|^2 - \vec{\Omega}\cdot\hat{L} \right)\psi_j.$$

(1)

Here we assume for simplicity that both components have the same mass $m$ and that rotation acts on each component equally. As we stir only along the $z$-axis with ro-

tation frequency $\vec{\Omega} = \Omega\vec{z}$, the angular momentum term becomes $\Omega L_z = -\hbar\Omega(x\partial_y - y\partial_x)$. The atom-atom interaction between the two components is described by $g_{12}$, and $g_{11}$ and $g_{22}$ represent the atom-atom interactions within components 1 and 2, respectively. The coupling constants $g_{ij} = \sqrt{8\pi}\hbar^2 a_{ij}/(ml_z)$ are dependent on the three dimensional scattering length $a_{ij}$, and the characteristic harmonic oscillator length in the $z$ direction, $l_z = \sqrt{\hbar/m\omega_z}$. We assume identical out-of plane trapping frequencies, $\omega_z$, for both components. For simplicity we also select the atom-atom interactions within each component to be the same ($a_{11} = a_{22} = a$ and $g_{11} = g_{22} = g$) and choose atom-atom interactions between the two components to be in the immiscible or phase-separated regime, that is $g_{12}^2 > g^2$ [23–25].

Phase-separation is driven by the need to minimise the interaction energy between the two-components, and immiscible condensates therefore assume a ground-state configuration that minimises the length of their adjoining domain wall. This means that immiscible two-component condensates in thin ring-geometries will phase separate azimuthally [5–7, 26], while radial phase separation is expected for wide traps when there is an imbalance in particle number in each condensate component [6, 26]. For elliptically shaped traps, the same interaction energy minimisation arguments follow and for the thinner trapping geometries we consider, azimuthal phase separation occurs exclusively [5, 26]. In racetrack geometries where the trap width is unequal along the minor and major elliptical axes, the two components will always phase-separate at the minimum azimuthal trapping width in order to minimise the interaction energy.

We impose superfluid flow around a racetrack by implementing an elliptical-shaped trapping potential, $V_j = V$ of the form [27]

$$V(r) = V_0 \cos^2(2\pi r/r_0) \tag{2}$$

where $r_0 = 2\pi/14 \times 10^4$ m, $r = \sqrt{(x/a)^2 + (y/b)^2}$ and we have chosen an ellipticity of $a/b = 1/1.8$, so that the racetrack potential traps more tightly along the minor elliptical axis. $V_0 = 8.66 \times 10^{-31}$ sets the trap depth and is chosen so that tunnelling is energetically unfavourable. The transverse trapping frequency is given by $\omega_z = 1000$ Hz and we model $0.75 \times 10^5$ Rb$^{87}$ atoms in each component on a grid of $1024^2$ points with spatial extent $-50\mu$m to $50\mu$m. We make use of the imaginary time propagation method (solving Eq. (1) using a split-operator spectral method [28] after a Wick rotation $t = -i\tau$) to find the initial ground state for the rotating two-component condensates with fixed $g$ and $g_{12}$. As expected, we find ground state configurations with the domain wall aligned along the minor elliptical axis (see Figs. 1(a) and 2(a)), which we use to investigate the subsequent real-time evolution dynamics in the laboratory frame.

In the following we will identify two flow regimes which are defined by the degree of immiscibility between the two condensate components or, alternatively, by the strength of rotation. They are characterised by very different su-

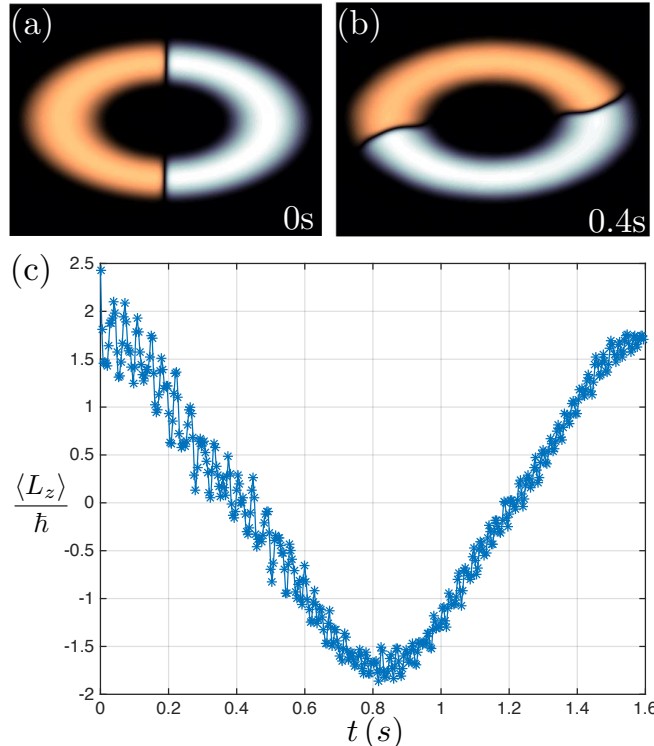

FIG. 1. Strongly phase-separated two component condensate ($g_{12} = 1.4g$) flowing around the racetrack with a rotation frequency $\Omega = 7$ Hz. (a) Density profiles of the two components (indicated by different colours) at $t = 0$, when $\langle L_z\rangle/\hbar$ is at a maximum and (b) at t=0.4 when it changes sign. Note that the phase boundary has been enhanced for clearer visibility. The time evolution of $\langle L_z\rangle/\hbar$ is depicted in (c).

perfluid flow behaviours, with one case displaying unidirectional and the other oscillating flow. Let us first discuss the oscillatory flow regime.

Similar to the rotationally symmetric taps, two-component currents in elliptically shaped trapping potentials display emergent solid body behaviour, with the phase boundary being required to adhere to classical solid body rotation [7]. As the condensates start flowing, the domain wall therefore moves towards the major elliptical axis where the trap strength is weaker, which requires it to grow in length. This growth directly translates to an increase in the inter-component interaction energy, which can be seen in Fig. 3 (circles), where the evolution of the different energy components is displayed (see the Appendix for details of the decomposition of the total energy). One can also directly see from this figure that the energy required to grow the domain wall comes at the expense of the rotational energy, $E_{\text{hyd}}$, which decreases as the domain wall flows towards the major elliptical axis and reaches almost zero around $t = 0.4$. However, as the domain wall has, at this time, not yet reached the major axis of the ellipse, the flow has to stop as no additional energy is available for the domain boundary to grow further.

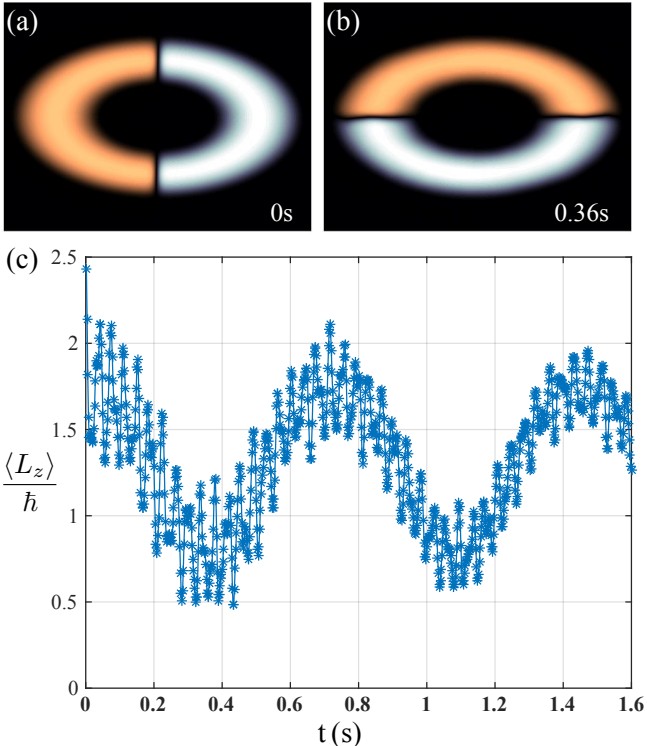

FIG. 2. Two component condensate flowing around the race-track, for $g_{12} = 1.2g$ and $\Omega = 7$ Hz. (a) Density profiles of the two components (indicated by different colours) at $t = 0$, when $\langle L_z \rangle / \hbar$ is at a maximum and (b) at t=0.36 when it is at a minimum. Note that the phase boundary has been enhanced for clearer visibility. The evolution of $\langle L_z \rangle / \hbar$ is depicted in (c).

This behaviour therefore corresponds to a decrease in the angular momentum per particle, $\langle L_z \rangle = i\hbar \int d\mathbf{x} \psi_j^* \left( y \frac{\partial}{\partial x} - x \frac{\partial}{\partial y} \right) \psi_j$, which is displayed in Fig. 1(c). In fact, one can see that at the points of zero angular momentum the direction of superfluid flow reverses, leading to sustained oscillations in the flow pattern [29]. The maximal magnitude of the average angular momentum corresponds to a minimum in the domain wall length, and occurs when the domain wall aligns with the minor elliptical axis (see Fig. 1(a) and Fig. 1(c)). The rotational, or hydrodynamic kinetic energy, oscillates out-of phase with the inter-component interaction energy, decreasing to zero when the direction of flow changes and increasing as the domain wall shortens (see Fig. 3(a)).

Careful examination of the different energies displayed in Fig. 3 shows that the intra-component interaction energy also oscillates out of phase with the inter-component interaction energy. In fact, it follows a behaviour similar to that of the rotational hydrodynamic energy and reaches a minimum when the domain wall is at its turning point (see Fig. 3(b)). This decrease in the intra-component interaction energy occurs due to the total

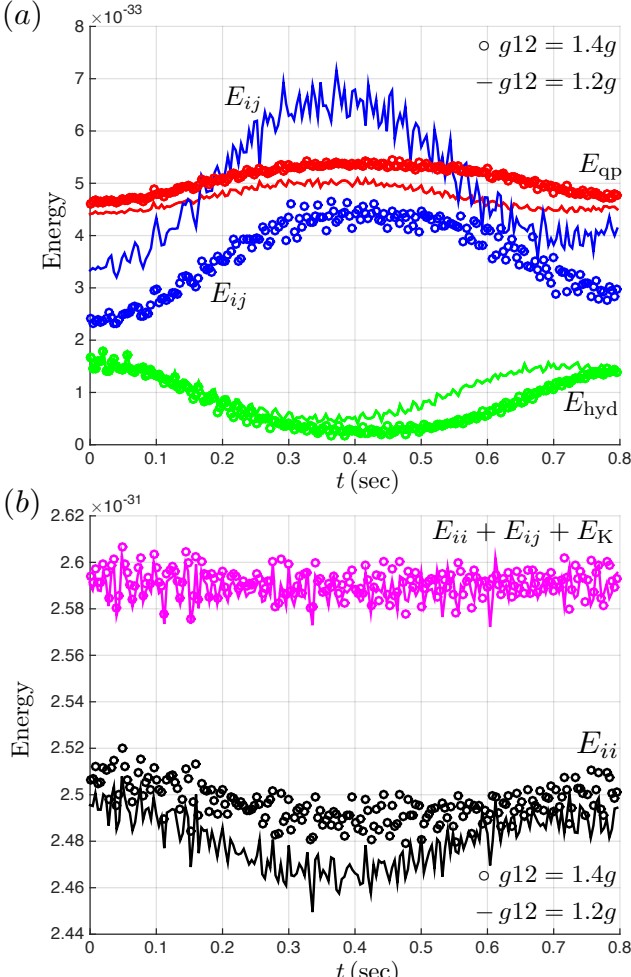

FIG. 3. Evolution of the different energy components for two component condensate superflow and superflow oscillations around the racetrack at an imposed rotation frequency ($\Omega = 7$ Hz) and for two inter-component interactions, $g_{12} = 1.2g$ and $g_{12} = 1.4g$ portrayed by lines and circles respectively. (a) Illustrates the growth of inter-component interactions, $E_{ij}$ (blue) and quantum pressure $E_{qp}$ (red) as the domain wall lengthens at the expense of the hydrodynamic kinetic energy $E_{hyd}$ (green) and intra-component interaction energy $E_{ii}$ ((black) shown in (b)). The total interaction and kinetic energy (magenta), shown in (b), is constant in time.

area of each component increasing because of an increased overlap region between the two components when the domain wall lengthens. As a result the area of maximum density decreases in the bulk of the components and the intra-component interaction energies decrease. The intra-component interaction energy can therefore also be used to grow the domain wall between the two components.

It is worth noting that the changes in the density distribution due to the lengthening of the domain wall also lead to an increase in the quantum pressure component of the kinetic energy, $E_{qp} = \int d\mathbf{x} \sum_j^{1,2} \frac{\hbar^2}{2m} |\nabla \sqrt{n_j}|^2$. Because the quantum pressure energy is purely dependent on the

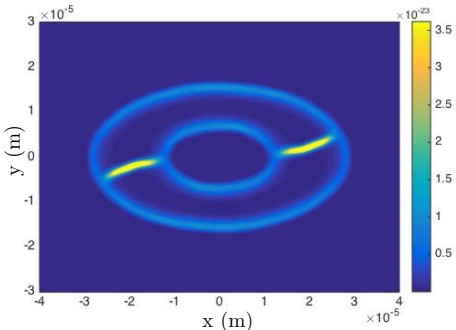

FIG. 4. The quantum pressure energy density, shown at $t = 0.4$s, clearly traces out both the domain wall and condensate edge. Illustrated for inter-component interactions, $g_{12} = 1.4g$, and a rotation frequency of $\Omega = 7$ Hz.

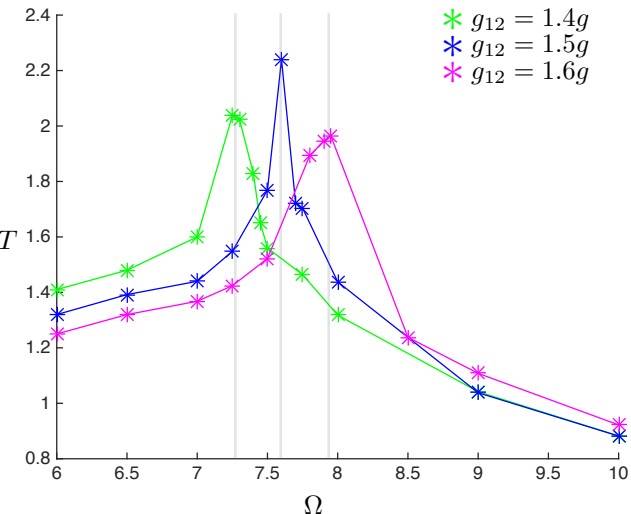

FIG. 5. The period taken for a complete oscillation or superflow around the racetrack increases closest to the transition point, and also shifts to larger external imposed rotation for stronger immiscibility. As the rotation strength is increased a transition to unidirectional superflow occurs at the longest period (peak). The lines indicating the peak are a guide to the eye.

gradient of the condensate density, its main contributions come from sharp variations in the condensate density, and consequently the quantum pressure energy density traces both the trap edge and the domain wall length between the two components as depicted in Fig. 4. As the domain wall length increases, the quantum pressure energy density therefore also increases (see Fig. 3(a)); however, these changes are much smaller than the variations in the interaction energy.

As the oscillation is mostly due to an interplay between the hydrodynamic and the inter-component energies, it is clear that changing either one of them can lead to a second dynamical regime, in which the superfluid flow becomes uni-directional. By increasing either the strength of external rotation beyond a critical value or re-

ducing the inter-component energies, the system can possess more hydrodynamic kinetic energy than the energy required to lengthen the domain wall to its maximum along the major elliptical axis. The system then shows unidirectional flow around the race-track potential, similar to phase-separated toroidally trapped condensates, where angular momentum must necessarily be conserved. However the flow slows down during this lengthening process and the average angular momentum per particle oscillates between two extrema with every $\pi/2$ rotation. As each of these extrema corresponds to an extremum in the domain wall length, one can immediately conclude that the rotational and the inter-component interaction energy of the system again oscillate out of phase (see Fig. 3, full lines): the rotational energy decreases in order to compensate for the increasing inter-component interaction energy, as the domain wall grows while moving towards the major elliptical axis (see Fig. 2(b) and Fig. 2(c)) and increases again as the domain wall length decreases as it traverses towards the minor elliptical axis.

In addition to oscillations in angular momentum that arise due to the growth and shrinking of the domain wall length, fast and small-scale oscillations are also evident in Figs. 1(c) and 2(c), which originate from excitations of the domain wall modes (see [29] and [30]). While in the rotationally isotropic case the system can compensate for the expected sheering of the domain wall due to azimuthal flow by creating a radial flow close to the phase boundary [7], this is not possible in a system where the curvature of the confining geometry changes. These excitations lead to the excitation of phonon modes in the condensate and a gradual decay in the slow-scale oscillations in angular momentum due to the changing domain wall length. We have confirmed that the frequency of these small-scale oscillations is independent of the rotation frequency and interaction strength and only depends on the confining geometry.

The transition point from oscillatory to uni-directional superflow as a function of the inter-component interaction strength and the rotation frequency can be deduced from the oscillation period, defined as the time taken for the domain wall to return to its initial position with the same direction of flow. This oscillation period is shown in Fig. 5, and one can see that the superfluid naturally slows down as the transition point from oscillating to unidirectional flow is approached.

While such an oscillation in the direction of superfluid flow can not be observed in toroidally trapped condensates as radial symmetry is preserved and enforces conservation of angular momentum, this effect is not only restricted to elliptical racetrack geometries. A change in superfluid velocity will be observable for immiscible multi-component superfluid flow in any geometry where the domain wall length must necessarily grow due to the confining geometry. One such example could be realised by slow multi-component superfluid flow along thin channels where the trap width increases along the channel, or a laval nozzle geometry, where the speed of flow along the

channel will decrease as the domain wall length increases [31–33].

In summary, we have presented an example of a multi-component superfluid system, where controlling the interplay between interaction and rotation can give rise to surprising superflow behaviour. In particular, we have explored superfluid flow around an elliptical race-track potential, which, in contrast to toroidal potentials, breaks both radial and azimuthal symmetry. Flow of immiscible superfluids around such a racetrack potential is no longer required to conserve angular momentum and exhibits unusual flow features. As we have demonstrated, in addition to traditional uni-directional superflow, tuning the rotational and interaction energies can lead to superfluid flow that oscillates in the direction of rotation, with the activation energy for this process being temporarily taken out of and restored into these energy components. Such an oscillatory superflow of immiscible condensates could be readily observed with current cold-atom experiments.

Acknowledgements: This work was supported by the Okinawa Institute of Science and Technology Graduate University and by JSPS KAKENHI-16K05461. We acknowledge discussions with Y. Hattori.

## I.  APPENDIX

In the laboratory frame, the total energy of our coupled condensate system can be written as a sum of the interaction $E_\mathrm{I}$, kinetic $E_\mathrm{K}$, and potential energies $E_\mathrm{V}$, $E_\mathrm{Tot} = E_\mathrm{I} + E_\mathrm{K} + E_\mathrm{V}$ which we write explicitly below. The interaction energy can be decomposed into the contributions from intra-component and inter-component in-

teractions, given by $E_{ii}$ and $E_{ij}$ respectively, as

$$E_{ii} = \frac{1}{2} \int d\mathbf{x} \left( g_{11}|\psi_1|^4 + g_{22}|\psi_2|^4 \right) \qquad (3)$$

$$\text{and } E_{ij} = \int d\mathbf{x} \left( g_{12}|\psi_1|^2|\psi_2|^2 \right) . \qquad (4)$$

The total kinetic energy, written as

$$E_\mathrm{K} = \int d\mathbf{x} \sum_j^{1,2} \frac{\hbar^2}{2m}|\nabla \psi_j|^2 , \qquad (5)$$

for the purpose of our work, is further decomposed into its quantum pressure contribution, and hydrodynamic kinetic energy contribution, the with the later containing incompressible (vortex) and compressible (sound) components. Writing the wave-function of each component in its Madelung representation, $\psi_j = \sqrt{n_j}\exp(i\phi_j)$, and recalling the condensate velocity for each component is $\mathbf{v}_j = (\hbar/m)\nabla\phi_j$, the kinetic energy becomes

$$E_\mathrm{K} = \int d\mathbf{x} \sum_j^{1,2} \frac{\hbar^2}{2m}|\nabla \sqrt{n_j}|^2 + \frac{\hbar^2}{2m}|\sqrt{n_j}\mathbf{v}_j|^2 , \qquad (6)$$

where the first term is the quantum pressure contribution, $E_\mathrm{qp}$, and the second term is the total hydrodynamic kinetic energy contribution, $E_\mathrm{hyd}$, to the kinetic energy. By identifying the divergence-free, $\nabla \cdot \left(\sqrt{n_j}\mathbf{v}_j\right)^i = 0$, and curl-free, $\nabla \times (\sqrt{n_j}\mathbf{v}_j)^c = 0$, components of the hydrodynamic kinetic energy contribution, we obtain the incompressible and compressible contributions to the hydrodynamic kinetic energy, respectively [34]:

$$E_\mathrm{hyd} = \int d\mathbf{x} \sum_j^{1,2} \frac{\hbar^2}{2m}|(\sqrt{n_j}\mathbf{v}_j)^i|^2 + \frac{\hbar^2}{2m}|(\sqrt{n_j}\mathbf{v}_j)^c|^2 . \quad (7)$$

Finally, we note the potential energy is expressed as

$$E_\mathrm{V} = \int d\mathbf{x} \sum_j^{1,2} V_j|\psi_j|^2 . \qquad (8)$$

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
