# Peer review of "Oscillating superflow in multicomponent Bose-Einstein condensates"

_SciPost Physics_

## Round 2 · Referee Report · Anonymous (Referee 1) · 2024-8-8

Report

The paper studies the real-time dynamics of immiscible binary Bose-Einstein condensates with an equal population in an elliptically shaped "racetrack" potential. The authors find two dynamical regimes of the flow dynamics by observing the motion of the domain walls, namely the oscillatory flow regime and the unidirectional flow regime. Using numerical simulations of the Gross-Pitaevskii equations, they show that the emergence of the two regimes depends on the external rotation frequency and the coupling strength between the components.

The authors demonstrated nontrivial nonlinear dynamics of a circular superflow although there have been some papers studying multicomponent Bose-Einstein condensates in an annular setup. The two dynamical regimes can be interpreted by the activation mechanism of the circular superflow, where the energy barrier may be now created by the intrinsic interaction properties of the binary Bose-Einstein condensates. In my opinion, the paper contains new results enough to be published. However, the underlying physics of the observed phenomena is a little bit hampered in the present version.

In Fig.5, the transition of the two dynamical regimes can be identified by looking at the oscillation period, which can give the critical value of the parameter. It is helpful to show additionally the phase diagram of the dynamics with respect to the rotational frequency and the intercomponent coupling.

In the current version, it is unclear whether such an energy barrier blocking the unidirectional superflow exists or not. If possible, it is useful to draw the adiabatic potential by changing the positions of the domain walls in the stationary configuration. If such an analysis is difficult, an even schematic diagram of the adiabatic potential would make it easier to understand the essence of the phenomena.

Recommendation

Ask for minor revision

---

## Round 2 · Referee Report · Anonymous (Referee 2) · 2024-8-20

Report

The present paper investigates the dynamics of an immiscible two-component Bose-Einstein condensate confined in an elliptic racetrack-shaped potential. The authors consider the initial state having an angular momentum and find two types of dynamics: oscillatory flow and uni-directional flow. The sign of the angular momentum is changed in the oscillatory flow, which is due to the energy transfer between the rotational kinetic energy and the interfacial interaction energy in the elliptic potential.

I think that the present paper contains interesting new results and recommend this paper for publication in SciPost Physics, after the following points are addressed.

1) I am confused about the direction of the rotation. Initially, the angular momentum of the system is positive, and therefore, the domain walls should start to rotate counterclockwise. However, Figs. 1(a), 1(b), 2(a), and 2(b) show that the pattern rotates clockwise.

2) In Figs. 1(a), 1(b), 2(a), and 2(b), the phase boundary is enhanced by black lines for clearer visibility. However, by the black line, it appears that the density becomes zero at the boundary, which is confusing.

3) The authors suppose rubidium atoms and assume a11 = a22. What hyperfine states do the two components correspond to? Also, the authors should state that the scattering length must be changed experimentally to realize g12 = 1.4 and 1.2 for the rubidium BEC.

Recommendation

Ask for minor revision

---

## Editorial Decision

awaiting_resubmission